# Drug-related problems among type 2 diabetic patients in Sunwal Municipality of Western Nepal

Sushma Chapagain[1], Nabin Pathak[2,3], Kushal Subedi[4], Prakash Ghimire[5], Bijay Adhikari[1], Niranjan Shrestha[1], Nim Bahadur Dangi[1]*

1 Pharmaceutical Sciences Program, Faculty of Health Sciences, School of Health and Allied Sciences, Pokhara University, Pokhara, Kaski, Nepal, 2 Drug Information Unit and Pharmacovigilance Cell, Hospital Pharmacy Department, Hetauda Hospital, Madan Bhandari Academy of Health Sciences, Hetauda, Makwanpur, Bagmati Province, Nepal, 3 Department of Pharmacy and Clinical Pharmacology, Madan Bhandari Academy of Health Sciences, Makwanpur, Hetauda, Bagmati Province, Nepal, 4 Pokhara University Teaching Hospital, Pokhara, Kaski, Nepal, 5 Universal College of Medical Sciences, Bhairahawa, Rupandehi, Nepal

* dcnim2@gmail.com

## Abstract

### Background

Several diseases co-exist with diabetes such as hypertension, and dyslipidemia, leading to cases of non-adherence, several drug interactions, and an increased risk of adverse drug reactions among patients, which are often termed as drug-related problems (DRPs). The role of pharmacists in high-income countries is well-defined in identifying DRPs among type 2 diabetes patients. However, these roles still need to be explored within low- and middle-income countries. The study aimed to identify DRPs in Type 2 diabetic patients.

### Methods

A community-based cross-sectional study was conducted in the Sunwal Municipality, Lumbini Province, Nepal from April to November 2021 where a stratified random sampling technique was employed to collect the data. The study included patients aged ≥ 18 years of either gender with type 2 diabetes who were prescribed at least one anti-diabetic medication. Patients were visited at their homes once identified through the community pharmacies, and a prescription review was conducted to identify the DRPs by using the Pharmaceutical Care Network Europe (PCNE) V8.02 tool and pertinent guidelines.

### Results

Among 182 patients, 97 (53.3%) had DRPs. Most of the patients were 50–60 years (n = 46; 25.3%), with a mean ± SD age of 55.43±14.46, as most were female (n = 94; 51.6%). Biguanides and sulfonylureas were the common classes of drugs prescribed. The major class of drug associated with DRPs were biguanides (n = 85; 49.7%), followed by sulfonylureas (n = 42; 24.6%). Metformin was the major drug associated with DRPs (n = 85; 49.4%). The major type of DRP identified was treatment effectiveness (n = 82; 79.61%), while patients

**Data Availability Statement:** All relevant data are within the manuscript and its Supporting information files.

**Funding:** The author(s) received no specific funding for this work.

not adhering to drug therapy (n = 97; 71.85%) was the leading cause of DRPs. DRPs were significantly associated with the duration of diabetes (p = .007) and the number of fruit servings (p = .007).

## Conclusion

The majority of the patients were found to have DRPs. The visiting patients at home by the pharmacists helped in identifying the DRPs and associated factors among type 2 diabetes patients, which may aid in the prevention and management of the disease.

## Introduction

The World Health Organization (WHO) defines non-communicable diseases (NCDs) as chronic conditions that persist over a long duration due to genetic, physiological, environmental, and behavioural factors. That accounts for approximately 77% of deaths in low and middle-income countries (LMICs) [1]. Diabetes, a major NCD, is a rapidly growing global health concern, with approximately 537 million people diagnosed in 2021. This number is expected to increase to 643 million by 2030 and 783 million by 2045, according to the International Diabetes Federation (IDF) Diabetes Atlas 10th edition report [2]. In South East Asia, the number is projected to reach 113 million by 2030 and 151 million by 2045, with its prevalence rising particularly in LMICs [3].

A family history of diabetes and young age are risk factors for type 1 diabetes [4], while lifestyle factors such as physical inactivity, sedentary lifestyle, obesity [5], and a low-fiber diet with a high glycemic index are risk factors for type 2 diabetes [6]. These factors accounts for over 90% of diabetes cases worldwide [2]. Diabetes often coexists with other conditions such as high blood pressure and hyperlipidemia and can lead to several complications including retinopathy, nephropathy, and neuropathy, often resulting in amputation [7].

Diabetic patients with comorbidities and complications often require multiple medications, leading to polypharmacy [8, 9], which further increases the likelihood of experiencing drug-related problems (DRPs) [10–12]. The Pharmaceutical Care Network Europe (PCNE) defines DRPs as events involving drug therapy that potentially interfere with desired health outcomes [13]. Several risk factors contribute to DRPs, including the comorbidities associated with diabetes [14, 15].

DRPs contribute to numerous adverse consequences. These can be enlisted as increased morbidity, mortality, higher health expenditure [16], emergency department visits, prolonged hospitalization, additional office visits, and long-term care admissions [17], highlighting the need for proper identification and timely intervention as it is a significant aspect of pharmaceutical care [16, 18]. Literature on the identification of DRPs among type 2 diabetes patients is rare within the Nepalese context, as most pharmacists are often confined to dispensing and procuring medicines within hospital settings [19].

Medication-related risk factors can occur within patients' homes. That can be because of poor adherence, medication storage issues with expired medication, and medication hoarding, leading to poor health outcomes [20]. Therefore, home visits can facilitate proper prescription reviews and assessment of patients' medication-taking behaviour, aiding in the identification of DRPs among type 2 diabetes patients [21]. In the Nepalese scenario, pharmacists visiting patients' homes to deliver pharmaceutical services is a new concept. Hence, this study aims to identify DRPs, assess different drug treatment measures, and explore the association between

socio-demographic variables, lifestyle factors, and DRPs by visiting type 2 diabetes patients in the community of Sunwal Municipality, Western Nepal.

## Methods

A community-based cross-sectional study was conducted in Sunwal Municipality, Lumbini Province, Nepal, from April to November 2021. Sunwal is one of the largest municipalities by population in the Nawalparasi district and serves as the urban core of a rapidly growing urban agglomeration in Nepal [22]. The study included patients aged ≥18 years of either gender diagnosed with type 2 diabetes and prescribed at least one anti-diabetic drug. Pregnant women, individuals with mental illness, and critically ill patients were excluded from the study. Ethical approval was obtained from the Institutional Review Committee (IRC) of Pokhara University, Nepal (Ref No 3-077-078). The sample size was calculated considering an 8.4% prevalence [23] and a 5% level of significance, with a design effect of 1.5, and assuming an infinite population using OpenEpi. The required minimum sample size (n) was found to be 178. However, the final data were collected from 182 diabetic patients.

Sunwal Municipality consists of 13 wards [22]. Therefore, a stratified random sampling technique was employed, where wards were considered as strata, and the sample was collected equally (i.e., 14) from all wards of the municipality. The patients were identified through community pharmacies and a community medical lab located within Sunwal Municipality. Those patients diagnosed with type 2 diabetes who visited these pharmacies or medical labs for refills, laboratory visits, or counseling purposes were identified. Once their contact details and addresses were obtained, patients were visited at their homes, and data was collected after obtaining written informed consent from them.

A structured patient profile form was used to collect data. That included sociodemographic parameters such as age, gender, body mass index, education, smoking habits, alcohol-drinking status, dietary patterns, and lifestyle. The physical activity was assessed using the Global Physical Activity Questionnaire [24], which helped in collecting information on physical activity participation in three settings: activity at work, travel to and from places, recreational activities, and sedentary behaviour.

The subjective, objective, assessment, and plan (SOAP) data were collected using SOAP notes based on the patient's recent prescription by the doctor. The American Diabetes Association (ADA) Standards of Medical Care in Diabetes 2015 [25] were used to assess the treatment effectiveness of the prescription. The validated tool PCNE classification version 8.02 was used to characterize the DRPs and their causes. The PCNE 8.02 tool, developed by the PCNE foundation, focuses on detecting DRPs, improving pharmaceutical health outcomes, and promoting pharmaceutical care. Here, PCNE 8.02 consists of separate domains and sub-domains for problems (3 primary domains and 7 sub-domains) and causes (8 primary domains and 35 sub-domains) [13].

Once the prescription review and complete SOAP analysis were conducted, DRPs were identified with the help of standard literature, textbooks, guidelines, and tertiary sources. Examples such as prescriptions not adhering to standard literature (ADA guidelines), out-of-range laboratory values, cases of non-compliance, or any patient-related factors were quantified under the problem domain. But the possible causes were determined depending on the observed problems. The problems and causes included were discussed with all the researchers and quantified only when all researchers reached a consensus. The data was analyzed using the Statistical Package for the Social Sciences (SPSS) version 16. Normally distributed results were expressed as mean ± standard deviation, while skewed data were presented as median with the

minimum and maximum values. The chi-squared test was used to test the association between DRPs and categorical variables, with statistical significance assumed at $p < 0.05$.

We categorized the DIET SERVINGS according to Oli et al., 2013 [26]. For fruits: One big apple, orange, and two pieces of banana were considered as 1 serving, while one small apple, orange, and one banana equated to ½ serving. Regarding vegetables: consuming vegetables to fill a bowl with a capacity of 200 ml was considered 1 serving, and filling half bowl with a capacity of 100 ml was considered as ½ serving.

BODY MASS INDEX (BMI) is calculated as the weight of a patient measured in kilograms divided by the square of their height in meters [27]. The BMI values were categorized as follows: BMI<18.5 indicates underweight, BMI 18.5–24.99 indicates normal weight, BMI 25–29.9 indicates overweight, and > 30–40 indicates obesity [28].

PHYSICAL ACTIVITY is classified based on metabolic equivalent units (MET) values. Less than 600 MET is considered low physical activity, 600–3000 MET is categorized as moderate, and above 3000 MET is considered high physical activity.

GLYCEMIC TARGETS include the assessment of glycemic control, and monitoring for diabetic patients according to ADA guidelines. Which stipulate levels of less than 130 mg/dl for fasting or pre-prandial glucose, 180 mg/dl for postprandial, and random blood glucose [25].

## Results

### Sociodemographic characteristics

A total of 182 diabetic patients were interviewed. The majority of the patients belonged to the age group 50–60 years (n = 46; 25.3%), with female patients (n = 94; 51.6%). The mean ± standard deviation of the age of patients was 55.43±14.46 years. The highest number of patients were unemployed (n = 53; 29.1%) and married (n = 164; 90.1%). In terms of ethnicity, Brahmins were the most prevalent (n = 61; 33.5%), followed by Chhetris (n = 39; 21.4%). Regarding educational status, 40 (22.0%) patients had a basic level of education. Sixty-four (35.2%) patients had a family history of diabetes, while 117 (64.3%) had comorbid conditions, with hypertension (n = 46; 25.3%) being the predominant one (Table 1). The distribution of anthropometric measurements showed that the majority were overweight (n = 85; 46.7%), followed by those with a normal BMI (n = 78; 42.9%), obese (n = 14; 7.7%), with the remaining classified as underweight (n = 5; 2.7%). The duration of diabetes varied among patients, with fifty-eight (31.9%) having a history of 5–10 years, seven (3.8%) recently diagnosed with type 2 diabetes, and thirty-six (19.8%) having a history exceeding 10 years.

### Major therapeutic class of drugs used by patients

The medications prescribed for diabetic patients were classified according to the Anatomical Therapeutic Chemical Classification system (ATC). The anatomical group, Alimentary tract and metabolism, consisted of 381 (56.19%) drugs, whereas cardiovascular system consisted of 182 (26.84%), genitourinary system and sex hormones consisted of 16 (2.36%), musculoskeletal system consisted of 25 (3.69%), and respiratory system consisted of 27 (3.69%). Within the alimentary tract group, biguanides (n = 164) were the most commonly used class of drugs, followed by sulfonylureas (n = 89) (Table 2).

### Lifestyle-Related characteristics of patients

The total of 182 diabetic patients, only one-third (n = 55; 30.2%) consumed fruits two days per week, while 52 (28.6%) consumed fruits one day per week. All the patients consumed half a

**Table 1. Socio-Demographic characteristics of patients.**

| Variable | n (%) |
|---|---|
| **Age (years)** | |
| 20–30 | 6 (3.3) |
| 30–40 | 31 (17.0) |
| 40–50 | 33 (18.1) |
| 50–60 | 46 (25.3) |
| 60–70 | 40 (22.0) |
| 70–80 | 19 (10.4) |
| >80 | 7 (3.8) |
| **Gender** | |
| Male | 88 (48.4) |
| Female | 94 (51.6) |
| **Occupation** | |
| Government employee | 18 (9.9) |
| Business | 35 (19.2) |
| Agriculture | 36 (19.8) |
| Unemployed | 53 (29.1) |
| Other | 40 (22.0) |
| **Marital status** | |
| Married | 164 (90.1) |
| Unmarried | 3 (1.6) |
| Widow | 15 (8.2) |
| **Ethnicity** | |
| Brahmin | 61 (33.5) |
| Chhetri | 39 (21.4) |
| Newar | 10 (5.5) |
| Janajati | 38 (20.9) |
| Dalit and others | 34 (18.7) |
| **Education** | |
| Illiterate | 30 (16.5) |
| Literate but informal education | 38 (20.9) |
| Primary level | 40 (22.0) |
| Secondary level | 35 (19.2) |
| Bachelor | 24 (13.2) |
| Master | 15 (8.2) |
| **Family History** | |
| Yes | 64 (35.2) |
| No | 118 (64.8) |
| **Comorbidities** | |
| Yes | 117 (64.3) |
| No | 65 (35.7) |
| If yes | |
| Hypertension | 46 (39.3) |
| Hypertension with others | 37 (31.6) |
| Others* | 34 (29.1) |

*Others: COPD, Asthma, Hyperlipidemia, Hypothyroidism, uric acid, Benign Prostate Hypertrophy, Rheumatic arthritis

**Table 2. Major therapeutic class of drugs used by patients.**

| ATC System | ATC Code | Medicine Mainly Used | No of Drugs | Total (n = 679) | Percent (%) |
|---|---|---|---|---|---|
| **Alimentary Tract and Metabolism** | A02B | Drugs for acid-related disorders | 24 | 381 | 56.12 |
| | A03FA | Propulsives | 1 | | |
| | A06AD | Osmotically acting laxative | 1 | | |
| | A10AB | Fast acting insulin | 1 | | |
| | A10AD | Combination insulin | 12 | | |
| | A10AE | Long-acting insulin | 2 | | |
| | A10BA | Biguanides | 164 | | |
| | A10 BB | Sulfonylureas | 89 | | |
| | A10BF | Alpha-glucosidase inhibitors | 21 | | |
| | A10BH | DPP-4 inhibitors | 51 | | |
| | A10BK | SGLT-2 inhibitors | 5 | | |
| | A11 | Vitamins | 7 | | |
| | A12 | Mineral supplements | 3 | | |
| **Blood and Blood Forming Organs** | B03 | Anti-anemic preparations | 12 | 12 | 1.77 |
| **Cardiovascular System** | C01AA | Cardiac glycosides | 2 | 182 | 26.80 |
| | C01DA | Organic nitrates | 1 | | |
| | C02C | Antiadrenergic agents (peripherally acting) | 5 | | |
| | C03 | Diuretics | 24 | | |
| | C05BX | Other sclerosing agents | 1 | | |
| | C07 | Beta-blocking agents | 29 | | |
| | C08 | Calcium channel blockers | 46 | | |
| | C09AA | ACE inhibitors | 3 | | |
| | C09CA | ARB | 45 | | |
| | C10A | Lipid modifying agents | 26 | | |
| **Genitourinary System and Sex Hormones** | G04 | Urologicals | 16 | 16 | 2.36 |
| **Systemic Hormonal Preparations (Thyroid only)** | H03AA | Thyroid hormones | 13 | 15 | 2.21 |
| | H03B | Antithyroid preparations | 2 | | |
| Anti-infective for Systemic Use | J01DD | Third generation cephalosporins | 1 | 3 | 0.44 |
| | J01CF | Beta-lactamase resistance penicillin | 1 | | |
| | J01XD | Imidazole derivatives | 1 | | |
| **Antineoplastic and Immunomodulating Agents** | L04AX | Other Immunosuppressants | 2 | 2 | 0.29 |
| **Musculo-skeletal System** | M01A | Anti-inflammatory and antirheumatic products, non-steroids | 13 | 25 | 3.68 |
| | M04 | Anti-gout preparations | 12 | | |
| **Nervous System** | N01B | Anesthetics, local | 1 | 14 | 2.06 |
| | N02 | Analgesics | 1 | | |
| | N03A | Antiepileptic | 7 | | |
| | N06A | Antidepressants | 5 | | |
| **Respiratory System** | R01 | Nasal Preparations | 2 | 27 | 3.98 |
| | R03 | Drug for obstructive airway diseases | 25 | | |
| **Various** | V03 | All other therapeutics products | 2 | 2 | 0.29 |

Anatomical System here refers to the system as classified by Norwegian Institute of Public Health WHO Collaborating Centre for Drug Statistics Methodology (https://atcddd.fhi.no/atc_ddd_index/).

ATC: Anatomical Therapeutic Chemical Classification, ACE: Angiotensin Converting enzyme, ARB: Angiotensin-II Receptor Blockers, DPP-4: Dipeptidyl Peptidase-4, SGLT-2: Sodium Glucose Transport Protein 2

**Table 3. Lifestyle-Related characteristics of patients.**

| Characteristics | n (%) |
|---|---|
| **Number of days the patient eats fruits in a typical week** | |
| 1 | 52 (28.6) |
| 2 | 55 (30.2) |
| 3 | 46 (25.3) |
| 4 | 19 (10.4) |
| 5 | 5 (2.7) |
| 6 | 2 (1.1) |
| 7 | 3 (1.7) |
| **Number of days the patient eats vegetables in a typical week** | |
| 3 | 8 (4.4) |
| 4 | 7 (3.8) |
| 5 | 29 (15.9) |
| 6 | 6 (3.3) |
| 7 | 132 (72.6) |
| **Smoking Status** | |
| Ex-smoker | 70 (38.4) |
| Current Smoker | 28 (15.4) |
| Non-smoker | 84 (46.2) |
| **Alcoholic Status** | |
| Ex-alcoholic | 54 (29.7) |
| Current alcoholic | 45 (24.7) |
| Non-alcoholic | 83 (45.6) |
| **Physical Activity** | |
| Low Physical activity | 89 (48.9) |
| Moderate Physical activity | 54 (29.7) |
| High Physical activity | 39 (21.4) |

serving of fruit on one of those days. Similarly, the majority (n = 132; 72.6%) of patients consumed vegetables seven days a week, while 29 (15.9%) consumed vegetables five days a week. Likewise, all the patients consumed two servings of vegetables on those days. The majority of patients were non-smokers (n = 84; 46.2%) and non-alcoholic (n = 83; 45.6%). Nearly half of the patients (n = 89; 48.9%) had low physical activity (Table 3).

## Drug Related Problems (DRPs)

Among 182 diabetic patients, 97 (53.3%) had DRPs. The major class of drugs associated with DRPs was Biguanides (n = 85; 49.7%), followed by Sulfonylureas (n = 42; 24.6%), and DPP-4 inhibitors (n = 24; 14.0%). Regarding single molecules, metformin was the major drug associated with DRPs (n = 85; 49.4%), followed by glimepiride (n = 37; 21.5%). Drugs belonging to the class of DPP-4 inhibitors, such as linagliptin (n = 14; 8.1%) and sitagliptin (n = 10; 5.8%), were also responsible for causing DRPs (Table 4).

The DRPs were categorized according to the PCNE V8.02 tool. Treatment effectiveness (P1) was the main type of DRP identified (n = 82; 79.61%), followed by the subdomain Others (P3) (n = 21; 20.39%). Within the primary domain of treatment effectiveness (P1), the effect of drug treatment not being optimal (P1.2) was the only problem identified. In the Others (P3) primary domain, unclear problem/complaint (P3.3) was the major type of problem identified (n = 20; 19.42%), followed by unnecessary drug treatment (P3.2) (n = 1; 0.97%). The effect of

**Table 4. Drugs class and drugs responsible for DRPs.**

| Characteristics | Number (n) | Percent (%) |
|---|---|---|
| **DRP** | | |
| Yes | 97 | 53.3 |
| No | 85 | 46.7 |
| **Drug Class associated with DRPs** | | |
| Biguanide | 85 | 49.7 |
| Sulfonylureas | 42 | 24.6 |
| Dipeptidyl peptidase-4 (DPP-4) inhibitors | 24 | 14.0 |
| Alpha glucosidase inhibitors | 9 | 5.2 |
| Insulin | 7 | 4.1 |
| Sodium-glucose Cotransporter- 2 (SGLT-2) inhibitors | 2 | 1.2 |
| Corticosteroid | 1 | 0.6 |
| Phosphodiesterase (PDE) inhibitor | 1 | 0.6 |
| **Drug associated with DRPs*** | | |
| Metformin HCL | 85 | 49.4 |
| Glimepiride | 37 | 21.5 |
| Sitagliptin | 10 | 5.8 |
| Linagliptin | 14 | 8.1 |
| Gliclazide | 5 | 2.9 |
| Voglibose | 4 | 2.3 |
| Insulin | 7 | 4.1 |
| Acarbose | 5 | 2.9 |
| Teneligliptin | 1 | 0.6 |
| Empagliflozin | 2 | 1.2 |
| Prednisolone | 1 | 0.6 |
| Sildenafil citrate | 1 | 0.6 |

* Multiple Responses

drug treatment not being optimal was assessed by monitoring the patient's lab values and SOAP format, which was later checked against the ADA guidelines. The unclear problem (P3.3) included issues with diet management, untreated conditions, inappropriate dose compliance by insulin users, and missed doses by older patients who had no caregiver (Table 5).

## Associated factors of DRPs

The primary domain cause, patient-related (C7.1), was found to be the major cause responsible for DRPs (n = 97; 71.85%), followed by the drug use process (C6.1) (n = 9; 6.67%). The dose

**Table 5. Drug-Related problems identified.**

| Primary Domain (Code) | Sub Domain (Problems with Code) | (n = 103,%) |
|---|---|---|
| Treatment effectiveness P1 | No effect of drug Treatment P1.1 | 0 (0) |
| | Effect of drug treatment not optimal P1.2 | 82 (79.61) |
| | Untreated symptoms or indication P1.3 | 0 (0) |
| Treatment Safety P2 | Adverse drug event (possibly) occurring P2.1 | 0 (0) |
| Others P3 | Problem with cost-effectiveness of the treatment P3.1 | 0 (0) |
| | Unnecessary drug-treatment P3.2 | 1 (0.97) |
| | Unclear problem/complaint P3.3 | 20 (19.42) |

**Table 6. Associated factors of DRPs.**

| Primary Domain (Code) | Sub Domain (Problems with Code) | n = 135, n (%) |
|---|---|---|
| Dose selection C3 | Drug dose too low C3.1 | 4 (2.96) |
| Dispensing C5 | Necessary information not provided C5.2 | 4 (2.96) |
| Drug use process C6 | Inappropriate timing of administration and/or dosing intervals C6.1 | 9 (6.67) |
| | Drug under-administered C6.2 | 3 (2.22) |
| | Drug over-administered C6.3 | 4 (2.96) |
| Patient Related C7 | Patient uses/takes less drug than prescribed or does not take the drug at all C7.1 | 97 (71.85) |
| | Patient uses/takes more drug than prescribed C7.2 | 1 (0.74) |
| | Patient uses unnecessary drug C7.4 | 1 (0.74) |
| | Patient stores drug inappropriately C7.6 | 1 (0.74) |
| | Inappropriate timing or dosing intervals C7.7 | 4 (2.96) |
| | Patient administers/uses the drug in a wrong way C7.8 | 6 (4.44) |
| | Patient unable to use drug/form as directed C7.9 | 1 (0.74) |

Note: We have only included and quantified the identified causes.

selection (C3) and dispensing (C5) categories remained the same (Table 6). The association between DRPs and independent variables revealed significant correlations with the duration of diabetes (p = .007) and fruit serving (p = .007) in bivariate analysis, while no significant associations were found with other independent variables (Table 7).

## Discussion

Among 182 patients, 103 DRPs were identified in 97 patients. Since there are few articles related to the prevalence of DRPs in communities. So, we compared the data with references from the hospital and community pharmacies based on the patients diagnosed with Diabetes mellitus. For instance, a study conducted in Hiwot Fana Specialized University Hospital, Harar, Eastern Ethiopia with type 2 diabetes with hypertension showed 364 DRPs (1.8 DRPs per patient) [29], while a total of 406 DRPs (1.9 average DRPs per patient) were identified in 191 patients in a retrospective study conducted in a tertiary hospital in Malaysia [30]. There is a wide difference in both studies due to the duration followed by the settings and the type of study. We conducted a cross-sectional analytical study using PCNE V8.02 to identify DRPs, which resulted in 53.3% of patients experiencing problems. To compare our findings, a prevalence of 58.2% DRPs was found in a study conducted among patients with diabetes attending ambulatory care, where patients had at least one DRP. However, the study conducted among ambulatory type 2 diabetes patients at *Madda Walabu* University Goba Referral Hospital showed a much higher prevalence of 88% among patients with at least one or more drug therapy problems [31].

The major problem identified in our study was the effect of drug treatment not being optimal (P1.2) which accounted for 79.61%, followed by Unclear Problem (P3.2) at 19.42%. A similar problem was identified in a study conducted at a specialized university hospital in Ethiopia, resulting in 49.2% of patients with drug treatment not being optimal [29]. About 80% of identified DRPs were related to the indication and effectiveness of therapy [32], while in a study conducted in Northern Cyprus, treatment effectiveness (22%) and unclear problems and complaints (19%) were observed [33]. This variation might have existed due to differences in study design, settings, populations, and the DRP classification tools used. For example,

**Table 7. Association of independent variable with DRPs.**

| Characteristics | DRPs | | Total (n = 182) | $\chi^2$ | p-value |
|---|---|---|---|---|---|
| | Yes (n, %) | No (n, %) | | | |
| **Age (years)** | | | | | |
| ≤55 | 41 (47.67) | 45 (52.33) | 86 | 2.071 | .150 |
| >55 | 56 (58.33) | 40 (41.67) | 96 | | |
| **Gender** | | | | | |
| Male | 47 (53.41) | 41 (46.59) | 88 | 0.001 | .977 |
| Female | 50 (53.19) | 44 (46.81) | 94 | | |
| **Marital status** | | | | | |
| Married | 84 (51.22) | 80 (48.78) | 164 | 2.874 | .090 |
| Others | 13 (72.22) | 5 (27.78) | 18 | | |
| **Ethnicity** | | | | | |
| Brahmin/Chhetri | 56 (56.00) | 44 (44.00) | 100 | 0.652 | .420 |
| Others | 41 (50.00) | 41 (50.00) | 82 | | |
| **Occupation** | | | | | |
| Government employee | 8 (44.44) | 10 (55.56) | 18 | 7.229 | .124 |
| Business | 21 (60.00) | 14 (40.00) | 35 | | |
| Agriculture | 13 (36.11) | 23 (63.89) | 36 | | |
| Unemployed | 33 (62.26) | 20 (37.74) | 53 | | |
| Others | 22 (55.00) | 18 (45.00) | 40 | | |
| **Education** | | | | | |
| Illiterate | 20 (66.67) | 10 (33.33) | 30 | 2.580 | .108 |
| Literate | 77 (50.66) | 75 (49.34) | 152 | | |
| **Complication** | | | | | |
| Yes | 63 (53.85) | 54 (46.15) | 117 | 0.40 | .842 |
| No | 34 (52.31) | 31 (47.69) | 65 | | |
| **Duration of diabetes#** | | | | | |
| Recently diagnosed | 0 (0.00) | 7 (100) | 7 | 14.156 | .007* |
| Less than 2 years | 14 (46.67) | 16 (53.33) | 30 | | |
| 2–5 year | 29 (56.86) | 22 (43.14) | 51 | | |
| 5–10 year | 30 (51.72) | 28 (48.28) | 58 | | |
| More than 10 years | 24 (66.67) | 12 (33.33) | 36 | | |
| **Family History** | | | | | |
| Yes | 33 (51.56) | 31 (48.44) | 64 | 0.119 | .730 |
| No | 64 (54.24) | 54 (45.76) | 118 | | |
| **Lifestyle Factors** | | | | | |
| **BMI** | | | | | |
| Underweight | 2 (40.00) | 3 (60.00) | 5 | 0.479 | .924 |
| Normal | 41 (52.56) | 37 (47.44) | 78 | | |
| Overweight | 46 (54.12) | 39 (45.88) | 85 | | |
| Obese | 8 (57.14) | 6 (42.86) | 14 | | |
| **Smoking status** | | | | | |
| Ex-smoker | 39 (55.71) | 31 (44.29) | 70 | 0.698 | .706 |
| Current smoker | 13 (46.43) | 15 (53.57) | 28 | | |
| Non smoker | 45 (53.57) | 39 (46.43) | 84 | | |
| **Alcoholic status** | | | | | |

(*Continued*)

**Table 7.** (Continued)

| Characteristics | DRPs | | Total (n = 182) | $\chi^2$ | p-value |
|---|---|---|---|---|---|
| | Yes (n, %) | No (n, %) | | | |
| Ex alcoholic | 30 (55.56) | 24 (44.44) | 54 | 0.185 | .912 |
| Current alcoholic | 24 (53.33) | 21 (46.67) | 45 | | |
| Non alcoholic | 43 (51.81) | 40 (48.19) | 83 | | |
| **Diet Fruits** | | | | | |
| ≤2 | 66 (61.68) | 41 (38.32) | 107 | 7.335 | .007* |
| >2 | 31 (41.33) | 44 (58.67) | 75 | | |
| **Diet vegetables** | | | | | |
| ≤6 | 26 (52.00) | 24 (48.00) | 50 | 0.047 | .829 |
| >6 | 71 (53.79) | 61 (46.21) | 132 | | |
| **Physical activity** | | | | | |
| Low | 52 (58.43) | 37 (51.57) | 89 | 3.304 | .192 |
| Moderate | 29 (53.70) | 25 (46.30) | 54 | | |
| High | 16 (41.02) | 23 (58.98) | 39 | | |

[#]Duration of diabetes refers to how long did the patient had diabetes. It is classified as recently diagnosed (who were recently diagnosed as per laboratory confirmation) and patient with diabetes for less than 2 years, 2–5 years, 5–10 years and more than 10 years.

* Significant at p < .05

Gokcekus et al. included four community pharmacies and used PCNE V6.2 for DRP classification [19], whereas Al-Taani et al. conducted a multicentered study with a modified DRP classification system [32].

In this study patients not adhering to the prescribed drug (C7.1) (n = 97; 71.85%) was the major cause for DRPs. Since this study was conducted during the presence of COVID-19 in various parts of Nepal, patients may probably have forgotten to take their medication as they were likely focused on other health-related preparations for their safety. A study reported that almost 90% of noncompliance was due to patient forgetting to take their medicines [30]. Regarding noncompliance among geriatric patients, it may be correlated with cases of polypharmacy and decreased cognitive abilities [34].

The nonadherence to diabetic medication leads to further complications, poor glycemic control, cases of morbidity and mortality along increased expenditures [35]. Such nonadherence could have been due to the patient's lack of information on timely refill of the medicines, forgetfulness, increased costs of medicine along with poor knowledge of the health insurance services. This underscores the need for proper intervention at the prescription stage or by a medical social worker in educating patients about the proper role of medication adherence and its effects on health outcomes. This study did not find any statistically significant association between age and the occurrence of DRPs. With increasing age, the body's physiological, and metabolic parameters tend to change and make people more susceptible to chronic disease conditions, metabolic abnormalities and adverse drug reactions (ADRs) as a result they are vulnerable to developing DRPs [36–38]. Despite a considerable number of DRPs in patients over 50 years of age, our study didn't find any significant association between DRPs and age. Similarly, ADRs were not commonly observed. This could be due to variations in the study design, as it involved visits to patients' homes. Additionally, COVID-19 was prevalent in various parts of Nepal at the time of data collection, which might have affected patients' medication-taking behavior, leading to non-compliance and thus impacting the categorization of DRPs. However, this paves the way for a robust study in myriad elderly patients exposed to

diabetes and other risk factors to check for DRPs and their outcomes. Similarly, a significant association between the duration of diabetes and DRPs (p = .007) was observed. Over time, patients with diabetes tend to develop macro and microvascular complications due to poor glycemic control, resulting in complications, comorbidities, and variations in pharmacokinetic-pharmacodynamic parameters, leading to changes in outcomes [39]. Likewise, polypharmacy would increase with comorbidities and complications [8, 9], resulting in an increased likelihood of acquiring DRPs [10–12]. DRPs were common in patients with chronic diabetes. The majority of the patients (16.48%) had DRPs who had a history of diabetes for 5–10 years. In this study, the most commonly prescribed medication was Metformin (35.6%), followed by glimepiride (13.9%), with the most commonly prescribed combination therapy being Metformin + Glimepiride (14.5%), which was similar to the study, indicating metformin as a major component of prescriptions followed by Sulfonylureas [39]. The use of medicines among the patients was found to be in accordance with the treatment guidelines, ADA 2015 standards of practice. The guideline also presents metformin as the first-line agent for treating type 2 diabetes [25].

The patients' BMI has no association with the occurrence of DRPs in our study. However, a study showed that patients with higher BMI were more likely to have DRPs than patients with a low BMI [40]. The median BMI of our study falls within the slightly overweight range, with a median value of 25.23. There was no association found between physical activity, alcohol consumption, and smoking status with DRPs in diabetic patients. Similarly, a study also showed no statistical association with smoking and alcohol status [30]. However, it is also hypothesized that physical activity leads to an increased rate of insulin-stimulated glucose disposal at a defined insulin dose [41], which may contribute to minimizing the occurrence of DRPs. The study also showed a statistically significant association between dietary fruit consumption and DRPs (p = .007). Despite this, we found that patients who consumed fewer servings of fruit had higher DRPs. This difference may be attributed to the type of fruit, sugar content, and glycemic load [42].

Existing literature also highlights the involvement of pharmacists in identifying and managing DRPs in diabetes [11, 43–46]. The study results highlighted the need for a multidisciplinary approach to managing diabetes to prevent further complications and DRPs. In which pharmaceutical care services include medication therapy management [47], identification of DRPs, managing adherence, educating patients, adhering to relevant guidelines, managing ADRs, improving compliance, and supporting patients [45]. The DRPs are often associated with increased costs, morbidity, and mortality, thus affecting patient outcomes. This underscores the significance of DRPs in diabetic patients, highlighting their impact on health outcomes and healthcare costs, as well as the role of pharmacists in identifying such problems and factors that would aid in their prevention and promote pharmaceutical care.

Here, the implications for clinical practice and policy development of this study introduce the role of pharmacists in providing pharmaceutical delivery services by visiting patients' homes. This helps in identifying DRPs and promotes the idea of quality use of medicines. The implications of lifestyle factors such as fruit and vegetable consumption, smoking status, alcohol consumption, and physical activity on diabetes management and the occurrence of DRPs should also be given high importance. It seems imperative to address practical strategies for healthcare providers to mitigate the impact of noncompliance and treatment ineffectiveness through tailored patient education, medication adherence support, along multidisciplinary care approaches. The local governments can hire pharmacists in this regard to fulfil the clinical role of promoting the safe and rational use of medications. The pharmacists could help in identifying the drug class and medications implicated in DRPs and assist patients in resolving

the associated issues. The regulatory bodies could contribute by developing guidelines and policies that favour the pharmacist's role in home-based pharmaceutical delivery services as well.

This study had some limitations. Firstly, since the research was conducted within a limited geographical area, it is quite challenging to generalize the findings to other geographic regions or patient populations. Secondly, there were methodological limitations, such as the cross-sectional design, which relies on self-reported data and lacks longitudinal follow-up. Additionally, the study lacked the interpretation of variables such as polypharmacy, financial aspects, psychological status, and history of drug side effects, which might have affected the prevalence of DRPs. Furthermore, the patients' lab values were taken from their latest checkups as the tests were not conducted during the study period. Although the interventions, their acceptance, and the status of the problem were not planned and documented according to PCNE V8.02. Despite the patient counselling and referral to a medical doctor were performed. In terms of statistical analysis, effect sizes, such as Cramer's V or Phi coefficient, were not reported in addition to p-values. Confidence intervals or measures of variability were also not included. Similarly, the sample size was very small, hence the study had limited power. This underscores the need for large population-based studies with an interventional model to assess patient outcomes, along with a multidisciplinary team within several local municipalities. Such studies would help generate more robust and generalizable results, address these limitations, and further explore the complex relationships between patient characteristics, treatment factors, and DRPs.

## Conclusions

In this study, DRP was observed among the majority of type 2 diabetic patients in the community of Sunwal Municipality. The most common problem identified was treatment effectiveness (P1), while patients not adhering to drug therapy (C7.1) was the main cause of DRPs. Significant aspects like duration of diabetes and dietary fruit intake were found to be associated with DRPs. Home visits by pharmacists were effective in detecting DRPs and their related factors in type 2 diabetes patients. Therefore, integrating pharmacists' services at the local level can help manage type 2 diabetes and associated factors.

## Supporting information

**S1 Dataset. Minimal data set.**
(XLSX)

## Acknowledgments

We would like to express our thanks to all the patients who helped us with providing their prescription information and to Sunwal Municipality for data collection approval.

## Author Contributions

**Conceptualization:** Sushma Chapagain, Niranjan Shrestha, Nim Bahadur Dangi.

**Data curation:** Sushma Chapagain, Nabin Pathak, Kushal Subedi, Bijay Adhikari, Niranjan Shrestha.

**Formal analysis:** Sushma Chapagain, Kushal Subedi, Bijay Adhikari, Nim Bahadur Dangi.

**Investigation:** Kushal Subedi, Prakash Ghimire.

**Methodology:** Sushma Chapagain, Niranjan Shrestha, Nim Bahadur Dangi.

**Project administration:** Nabin Pathak, Prakash Ghimire, Nim Bahadur Dangi.

**Resources:** Nabin Pathak, Prakash Ghimire, Bijay Adhikari.

**Software:** Kushal Subedi, Bijay Adhikari.

**Supervision:** Niranjan Shrestha, Nim Bahadur Dangi.

**Validation:** Nabin Pathak, Kushal Subedi, Prakash Ghimire.

**Visualization:** Kushal Subedi, Prakash Ghimire.

**Writing – original draft:** Sushma Chapagain, Nabin Pathak, Nim Bahadur Dangi.

**Writing – review & editing:** Niranjan Shrestha, Nim Bahadur Dangi.

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
