## [Decision Letter · Decision Letter 0]

19 Mar 2024

PONE-D-23-23137Drug-Related Problems in Diabetic Patients in the Community of Sunwal Municipality, Western NepalPLOS ONE

Dear Dr. Nim Bahadur Dangi,

Thank you for submitting your manuscript to PLOS ONE. After careful consideration, we feel that it has merit but does not fully meet PLOS ONE’s publication criteria as it currently stands. Therefore, we invite you to submit a revised version of the manuscript that addresses the points raised during the review process.

We look forward to receiving your revised manuscript.

Kind regards,

Naeem Mubarak, PhD

Academic Editor

PLOS ONE

Journal Requirements:

4. Please ensure that you include a title page within your main document. You should list all authors and all affiliations as per our author instructions and clearly indicate the corresponding author.

5. We are unable to open your Supporting Information file Minimal Data Set.sav. Please kindly revise as necessary and re-upload.

Additional Editor Comments:

The article holds a great deal of merit for publication. However major revisions are necessary to further improve the quality and essence of the manuscript. The discussion lacks depth, implications and future prospects. To further improve the discussion section and highlight the role of pharmacists in the management of diabetes related problems, you may cite the following study (This is optional and should be taken as a suggestion for the improvement of the manuscript):

https://doi.org/10.3389/fpubh.2024.1323102

Reviewers' comments:

Reviewer's Responses to Questions

**Comments to the Author**

1. Is the manuscript technically sound, and do the data support the conclusions?

Reviewer #1: Partly

Reviewer #2: Partly

2. Has the statistical analysis been performed appropriately and rigorously? 

Reviewer #1: No

Reviewer #2: Yes

3. Have the authors made all data underlying the findings in their manuscript fully available?

Reviewer #1: Yes

Reviewer #2: No

4. Is the manuscript presented in an intelligible fashion and written in standard English?

Reviewer #1: No

Reviewer #2: No

5. Review Comments to the Author

Reviewer #1: The title lacks specificity regarding the focus of the study. Consider revising it to indicate that the research is about identifying drug-related problems in Type 2 diabetic patients in the Sunwal Municipality, Western Nepal.

The location information in the title is helpful for context, but it could be integrated more seamlessly into the title to improve readability. For example, consider rephrasing as "Drug-related problems in Type 2 diabetic patients in Sunwal Municipality, Western Nepal".

The abstract provides a concise study overview, highlighting the importance of identifying drug-related problems (DRPs) in Type 2 diabetic mellitus (T2DM) patients. However, it lacks specific details regarding the methodology and key findings, making it challenging for readers to grasp the significance of the research at a glance. Consider briefly summarising the methodology and the main results to enhance clarity and comprehensiveness.

The abstract provides a concise overview of the study's objectives and findings but could benefit from a more structured approach. Consider organizing it into sections (e.g., Background, Methods, Results, Conclusion) to improve clarity and readability.

In the introduction, you effectively outline the significance of non-communicable diseases (NCDs) and diabetes, providing relevant statistics and background information. However, the introduction could be improved by clearly stating the research gap or problem the study addresses. Additionally, while you discuss risk factors for diabetes, it would be beneficial to mention the relevance of these factors to the study's focus on DRPs in T2DM patients.

In the background section, provide a clearer transition from discussing non-communicable diseases in general to focusing specifically on diabetes.

Consider including a statement about the study's specific objectives in the introduction to clarify the research focus.

The methods section describes the study design, sampling technique, data collection tools, and analysis procedures. However, clarifying how pharmacists selected patients for home visits and the criteria used to identify DRPs would be helpful. Providing more information on the validation process of the data collection tools and reliability measures would enhance the rigour of the study.

Operational definitions are provided for key terms such as diet servings, BMI, physical activity, and glycemic targets, which are essential for standardizing data collection and analysis. However, the definition of glycemic targets could be expanded to include a brief rationale for monitoring these targets in diabetic patients according to ADA guidelines.

In the methods section, provide more details on how pharmacists selected patients for home visits, including any criteria used to prioritize visits or ensure representativeness.

Clarify the process of identifying drug-related problems and the specific criteria used for classification. Providing examples of the types of the issues identified would enhance understanding.

Include information on the reliability and validity of the data collection tools used in the study, such as the patient profile form and PCNE classification tool.

In the results section, consider including confidence intervals or measures of variability to provide a more comprehensive understanding of the data.

The discussion section effectively integrates the study findings with existing literature, but it could be strengthened by discussing potential implications for clinical practice and policy development.

Address potential limitations of the study, such as sampling bias or generalizability issues, and propose strategies to overcome these limitations in future research.

The results section provides a comprehensive overview of the sociodemographic characteristics, duration of diabetes, major therapeutic classes of drugs used, distribution of anthropometric measurements, lifestyle-related characteristics, drug-related problems (DRPs), and their causes.

The presentation of results in tabular format enhances readability and facilitates understanding. However, more context or interpretation should be provided for each table.

While the sociodemographic characteristics are well-described, discussing any notable trends or patterns observed within these demographics would be beneficial. For example, are there any significant differences in DRPs based on age, gender, occupation, or educational status?

The categorization of DRPs and their causes using the PCNE V8.02 tool is appropriate, but it would be helpful to provide a brief explanation of this tool for readers who may not be familiar with it.

When discussing the major therapeutic classes of drugs used by patients, elaborate on why certain classes are more prevalent and whether this aligns with current treatment guidelines or prescribing practices.

In the section on lifestyle-related characteristics, the findings regarding fruit and vegetable consumption, smoking status, alcohol consumption, and physical activity provide valuable insights. However, the implications of these lifestyle factors on diabetes management and the occurrence of DRPs should be considered.

When presenting the results related to DRPs, provide more context on the significance of specific drug classes and medications implicated in causing DRPs. Additionally, discuss any potential implications for clinical practice and patient care.

The association analysis between DRPs and sociodemographic variables is informative, but the interpretation could be enhanced by discussing possible reasons for the observed associations or lack thereof. For instance, why might the duration of diabetes and fruit serving be associated with DRPs while other variables are not?

In the discussion section, build upon the results by synthesizing findings from the literature and offering insights into the broader implications for diabetes management and pharmaceutical care practices in similar settings.

Overall, the results section provides valuable data but could benefit from deeper analysis and interpretation to enhance the understanding of the findings and their relevance to the study objectives.

The statistical analysis performed in this study appears to be appropriate for the research objectives outlined. However, there are a few areas where additional clarification or depth could enhance the interpretation of the findings.

Consider providing more information about the statistical tests used, particularly regarding the choice of the chi-square test for analyzing associations between categorical variables. Explaining why this test was chosen and whether its assumptions were met would be beneficial.

Provide details on the significance level (alpha) used for hypothesis testing. Mentioning the significance level (typically set at 0.05) would help readers understand the criteria for determining statistical significance.

When discussing the results of the chi-square tests, consider reporting the effect sizes, such as Cramer's V or Phi coefficient, in addition to p-values. Effect sizes provide valuable information about the strength and practical significance of the associations observed.

Evaluate the statistical tests' power to ensure that the sample size was sufficient to detect meaningful associations. If the study had limited power, acknowledge this limitation and discuss its potential impact on the interpretation of the results.

Discuss any potential confounding variables controlled for in the analysis or factors considered but not included in the final models. This information would help readers understand the robustness of the findings and the extent to which alternative explanations were ruled out.

Consider performing subgroup analyses or stratifying the results by relevant variables to explore potential effect modifiers or identify subgroup differences in the associations examined.

If applicable, discuss any adjustments made for multiple comparisons to control the overall Type I error rate. This is particularly important if multiple comparisons were conducted without appropriate adjustments, as it could inflate the likelihood of false positives.

Provide information on missing data handling and any sensitivity analyses conducted to assess the robustness of the findings in the presence of missing data or outliers.

Finally, the clinical significance of the statistically significant associations observed will be discussed. Consider whether the magnitude of the associations is meaningful in practice and how these findings may inform patient care or healthcare policies in the studied population.

Discussion

The discussion provides a comprehensive overview of the study findings but could benefit from clearer connections between the results and their broader implications. Consider explicitly linking each finding to its relevance for clinical practice, public health, or future research directions. Additionally, provide more context for the significance of DRPs in diabetic patients, emphasizing their impact on health outcomes and healthcare costs.

While the discussion compares the study findings with existing literature, it could be strengthened by a more critical examination of the similarities and differences between this study and previous research. Discuss potential reasons for discrepancies in prevalence rates, such as variations in study methodologies, populations, or healthcare systems. Highlighting areas of agreement and disagreement with prior studies would enhance the understanding of the current findings.

When discussing the associations between sociodemographic and lifestyle factors with DRPs, delve deeper into the potential mechanisms underlying these relationships. For example, explore why certain factors like duration of diabetes or dietary habits might be associated with a higher likelihood of experiencing DRPs. Providing theoretical explanations or hypotheses based on existing literature would enrich the interpretation of the statistical associations observed.

Also add how can pharmacist play role. You can cite this study from Nepal. (This is optional and should be taken as a suggestion for the improvement of the manuscript)

Sapkota B, Bokati P, Dangal S, Aryal P, Shrestha S. Initiation of the pharmacist-delivered antidiabetic medication therapy management services in a tertiary care hospital in Nepal. Medicine (Baltimore). 2022 Apr 22;101(16):e29192. doi: 10.1097/MD.0000000000029192. PMID: 35482989; PMCID: PMC9276257.

Expand on the clinical implications of the study findings beyond the identification of DRPs. Discuss how these findings could inform interventions to reduce DRPs and improve patient outcomes in diabetic populations. Consider addressing practical strategies for healthcare providers to mitigate the impact of noncompliance and treatment ineffectiveness, such as tailored patient education, medication adherence support, or multidisciplinary care approaches.

Limitations and Future Directions

While the discussion briefly mentions limitations related to the study design and population, it could be more comprehensive in acknowledging potential sources of bias or uncertainty. Discuss methodological limitations in detail, such as the cross-sectional design, reliance on self-reported data, or lack of longitudinal follow-up. Additionally, propose avenues for future research to address these limitations and further explore the complex relationships between patient characteristics, treatment factors, and DRPs.

Recognize the generalizability limitations inherent in the study's specific context and population. Discuss the implications of conducting the research in a community setting within Sunwal Municipality and acknowledge that the findings may not be applicable to other geographic regions or patient populations. Consider discussing the transferability of findings to similar settings or populations and provide suggestions for validating the results in diverse healthcare contexts.

Proofread the manuscript for grammatical errors and ensure consistency in formatting and style throughout the document.

Reviewer #2: 1. The World Health Organization (WHO) defines Non-communicable diseases (NCDs) as chronic diseases which occur for a longer duration of time as a consequence of genetic, physiological, environmental, and behavioral factors and is responsible for about 77% of death in low and middle income countries (LMIC). It appears as if the definition of NCD requires the death figures. To avoid misinterpretation. Split the statement.

2. Replace Familial history by family history and type 2 diabetic mellitus by type 2 diabetes mellitusthroughout.

3. ‘Pretesting was done 95 at 10% of the study population and reliability was calculated using Cronbach Alpha. The Cronbach alpha for patient profile, diet, and physical activity was found to be 0.701 while it was 0.706 for DRPs’: The statement is not clear and needs to be evaluated by and expert statistician. Cronbach’s alpha is a measure of internal consistency in a set of items used for a questionnaire. Authors should mention how and on which specific items Cronbach alpha was calculated. If not sure, the statement should be deleted.

4. Lines 114-115: Glycaemic targets: Should be less than and not above the values given.

5. Table 3: Authors should correct the Anatomical class (WHO-ATC). For example: Alimentary canal should be replaced by Alimentary tract and metabolism. Likewise blood should be replaced by blood and blood forming organs.

6. In coloumn 3 of Table 3, Authors have randomly used Therapeutic class for some drugs and Chemical subgroup. Please keep uniformity and use either of them and not both. Common individual drugs could have been mentioned example metformin, telmisartan etc.

7. Table 8: Authors can only mention the PCNE Class to which DRPs were observed rather than mentioning each cause

8. Table 9: Authors should revise the table. Give the total number in each group and the number (%) developing DRPs. For example 86 of 55 years and above among whom 41(47.7%) developed DRPs. This will ease the interpretation of data for the readers.

9. Discussion: Needs a thorough revision. Not comprehensible in the present form.

10. Lines 199-202:Mention rates of DRPs in the studies cited (28 and 29 references) in %

11. Lines 205-207: 58.2% is not much higher than 53.3% rates of DRPs observed in the present study. Correct the statement. Yes it is higher than 88% rates of DRPs mentioned in the next statement by authors.

12. What were the major DRPs in < 60 years (adults) and ≥ 60 years of age.

13. Since a considerable percentage of participants were older adults (50 years of age and above ), Authors should discuss other studies on Drug related problems in older adults.

14. Contrary to the findings of the present study, one large study (mentioned below )has shown hypoglycemia due to antiDM drugs as the commonest metabolic abnormality also the second common DRP causing hospitalization in older patients. 

Kaur U, Chakrabarti SS, Gupta GK, Singh A, Gambhir IS. Drug-related problems in older adults in outpatient settings: Results from a 6-year long prospective study in a tertiary hospital of north India. Geriatr Gerontol Int. 2023 Aug 14. doi: 10.1111/ggi.14650. Epub ahead of print. PMID: 37577765 (This is optional and should be taken as a suggestion for the improvement of the manuscript).  

15. Lines 242-243: In our study the most common comorbid condition associated with T2DM was hypertension. Similar result was obtained in the study conducted by Huri & Ling, 2013[29] and Ogbonna et al, 2014: Should be deleted. Discussion should focus on relevant findings of the study and not on co-morbidities or baseline characteristics unless they are determinants of DRPs

16. Lines 249- 251: “. In this study oral hypoglycemic agents (OHA) and insulin contain (n=275; 55.5%) of prescription as 70.6 as monotherapy, (24.1%) combination and insulin (5.4%)”: Not clear. Please Clarify

17. Line: 244: There is a statistical association between the duration of diabetes and

the DRPs (p=.007). What does this signify? DRPs common in recently diagnosed or in patients with chronic diabetes.

18. One important variable which authors missed was number of drugs taken concomitantly.

Polypharmacy shares a direct relation with DRPs such as adverse drug reactions.

19. Likewise financial status and psychological factors, history of side effects are important factors deciding compliance to therapy and were missed.

20. Lines 263-264: Patients eating less fruits were found to develop DRPs commonly. Authrs should elaborate the association found between duration of diabetes and fruits consumption and its significance

21. Lines 273: Interventions were not planned and carried out while the patient counseling and referral to medical doctor was done.: Statement is not clear. Were any interventions planned at pharmacist level after diagnosing DRPs?

22. Higher female enrolment does not mean higher % of females developing DRPs. Lines 230-235 should be modified.

23. Write a separate Limitation section before Conclusion

24. Modify the Conclusion part accordingly. Avoid statistical term, unclear problems etc. in conclusion and give only clinically relevant findings and factors with important logical associations with DRPs

25. Authors should discuss only important findings in the Discussion and not each finding of Results. For ex : Lines 239-243, 244-253 seem redundant

26. That non compliance was the commonest cause of DRP should be followed by a statement on the need of proper intervention at the prescription stage or by a medical social worker for importance of treatment adherence. Important reasons for non-compliance should also be sought in future.

27. Any reason why other more common DRPs such as hypoglycaemia, hypotension and hypertension were not observed

28. Tables 2 and 4 can be deleted as data is there in text.

6. PLOS authors have the option to publish the peer review history of their article (what does this mean?). If published, this will include your full peer review and any attached files.

Reviewer #1: No

Reviewer #2: No

---

## [Author Response · Author response to Decision Letter 0]

4 May 2024

Warm greetings!

We thank for the review of the paper and for the critical comments that have helped us to address the irregularities within our paper. We have revised the paper following your advice and the reviewer’s comment. In this file, we present the changes made with the line numbers to the respective comments made.

Editors’ Comments

• Please ensure that you include a title page within your main document. You should list all authors and all affiliations as per our author instructions and clearly indicate the corresponding author.

We have included a title page within our main document with all authors' lists and affiliations.

• We are unable to open your Supporting Information file Minimal Data Set.sav. Please kindly revise as necessary and re-upload.

Thank you for underlining this deficiency. We have now included the supporting information file in another (Excel) format and reuploaded it.

Additional Editor Comments:

The article holds a great deal of merit for publication. However major revisions are necessary to further improve the quality and essence of the manuscript. The discussion lacks depth, implications and future prospects. To further improve the discussion section and highlight the role of pharmacists in the management of diabetes related problems, you may cite the following study (This is optional and should be taken as a suggestion for the improvement of 

the manuscript):https://doi.org/10.3389/fpubh.2024.1323102

Response: Thank you for the general positive assessment of the paper. We have incorporated the reviewer’s comments and made amendments and additions to the contents of the paper. We have also included the role of the pharmacist in diabetes and in managing drug-related problems within our discussion section. Thank you, we have also cited the above doi link (reference number 46) to the manuscript. [Addressed in line numbers 234-351]

Reviewer #1: Comments

The title lacks specificity regarding the focus of the study. Consider revising it to indicate that the research is about identifying drug-related problems in Type 2 diabetic patients in the Sunwal Municipality, Western Nepal.

Thank you for underlining this deficiency. We have revised the title considering the review’s comments as “Drug-related Problems in Type 2 Diabetic Patients in Sunwal Municipality, Western Nepal.”

The location information in the title is helpful for context, but it could be integrated more seamlessly into the title to improve readability. For example, consider rephrasing as "Drug-related problems in Type 2 diabetic patients in Sunwal Municipality, Western Nepal".

We have now considered the reviewer’s comments and revised the title to “Drug-related Problems in type 2 Diabetic Patients in Sunwal Municipality, Western Nepal.”

The abstract provides a concise study overview, highlighting the importance of identifying drug-related problems (DRPs) in Type 2 diabetic mellitus (T2DM) patients. However, it lacks specific details regarding the methodology and key findings, making it challenging for readers to grasp the significance of the research at a glance. Consider briefly summarising the methodology and the main results to enhance clarity and comprehensiveness.

Thank you for underlining the deficiency. We have now made changes to the abstract section trying to make it brief.

The abstract provides a concise overview of the study's objectives and findings but could benefit from a more structured approach. Consider organizing it into sections (e.g., Background, Methods, Results, Conclusion) to improve clarity and readability.

Thank you for underlining the deficiency. We have separated the sections (Background, Methods, Results, and Conclusion) as suggested. 

In the introduction, you effectively outline the significance of non-communicable diseases (NCDs) and diabetes, providing relevant statistics and background information. However, the introduction could be improved by clearly stating the research gap or problem the study addresses. Additionally, while you discuss risk factors for diabetes, it would be beneficial to mention the relevance of these factors to the study's focus on DRPs in T2DM patients.

Thank you for underlining the deficiency. We have now mentioned and included the gap along with the introduction of risk factors of DRPs. [Addressed in Line numbers 78-80, 84-87, 92-96] 

In the background section, provide a clearer transition from discussing non-communicable diseases in general to focusing specifically on diabetes.

Thank you for the constructive feedback. We have now addressed the comments and made a clearer transition from discussing non-communicable diseases to diabetes. [Addressed in Line numbers 60-62, 62-67]

Consider including a statement about the study's specific objectives in the introduction to clarify the research focus.

Thank you for underlining the deficiency. We have specified the objective as suggested.

The methods section describes the study design, sampling technique, data collection tools, and analysis procedures. However, clarifying how pharmacists selected patients for home visits and the criteria used to identify DRPs would be helpful. Providing more information on the validation process of the data collection tools and reliability measures would enhance the rigour of the study.

Thank for addressing these issues. We have now added contents related to how home visits were conducted, how the DRPs were identified with information on validation. [Addressed in Line numbers 113-116, 126-130, 131-137]

Operational definitions are provided for key terms such as diet servings, BMI, physical activity, and glycemic targets, which are essential for standardizing data collection and analysis. However, the definition of glycemic targets could be expanded to include a brief rationale for monitoring these targets in diabetic patients according to ADA guidelines.

Thank you for highlighting this deficiency. We have now incorporated all information regarding glycemic targets. We have removed the heading "operational definition" and included this information at the end of the methodology section.

In the methods section, provide more details on how pharmacists selected patients for home visits, including any criteria used to prioritize visits or ensure representativeness.

Thank for addressing this issue. We have now added information related to this context. [Addressed in Line numbers 112-116]

Clarify the process of identifying drug-related problems and the specific criteria used for classification. Providing examples of the types of the issues identified would enhance understanding.

Thank you for pointing out the deficiency. We used PCNE V8.02 to classify the drug-related problems. We have now included information relevant to this context. As for the types of issues identified, we have incorporated them within the results section. 

Include information on the reliability and validity of the data collection tools used in the study, such as the patient profile form and PCNE classification tool.

PCNE V8.02 is a validated tool used to measure DRPs. We have now added information validity of the data collection tools used in the study such as PCNE classification tool. [Addressed in Line numbers 126-128]

In the results section, consider including confidence intervals or measures of variability to provide a more comprehensive understanding of the data.

Thank you for the comment. With due respect we want to inform the reviewer that we did not include confidence intervals or measures of variability. We have included this in our limitation section too. [Addressed in Line numbers 345-356]

The discussion section effectively integrates the study findings with existing literature, but it could be strengthened by discussing potential implications for clinical practice and policy development.

Thank you for highlighting this deficiency. We have now included a section on potential implications for clinical practice and policy development in the second last paragraph of the discussion section. However, we have not provided a separate heading, as the journal format does not include one. [Addressed in Line numbers 322-334]

Address potential limitations of the study, such as sampling bias or generalizability issues, and propose strategies to overcome these limitations in future research.

Thank you for underlining the deficiency. We have now included a section relating to strength and limitation of the study at the last paragraph of discussion section. [Addressed in Line numbers 335-351]

The results section provides a comprehensive overview of the sociodemographic characteristics, duration of diabetes, major therapeutic classes of drugs used, distribution of anthropometric measurements, lifestyle-related characteristics, drug-related problems (DRPs), and their causes.

Thank you for the general positive assessment of the result section. 

The presentation of results in tabular format enhances readability and facilitates understanding. However, more context or interpretation should be provided for each table.

Following the reviewer’s comment, we have added context to the table with short interpretations. We hope this justifies the reviewer’s comment.

While the sociodemographic characteristics are well-described, discussing any notable trends or patterns observed within these demographics would be beneficial. For example, are there any significant differences in DRPs based on age, gender, occupation, or educational status?

Thank you for the comment. No significant differences in DRPs based on age, gender, occupation, or educational status was observed. 

The categorization of DRPs and their causes using the PCNE V8.02 tool is appropriate, but it would be helpful to provide a brief explanation of this tool for readers who may not be familiar with it.

After the reviewer’s comment, we have now added content relating to a brief explanation of the tool. [Addressed in Line numbers 126-130]

When discussing the major therapeutic classes of drugs used by patients, elaborate on why certain classes are more prevalent and whether this aligns with current treatment guidelines or prescribing practices.

Thank you for underlining this deficiency. We have now added context regarding the use of drugs and its alignment with treatment guidelines. [Addressed in Line numbers 290-292]

In the section on lifestyle-related characteristics, the findings regarding fruit and vegetable consumption, smoking status, alcohol consumption, and physical activity provide valuable insights. However, the implications of these lifestyle factors on diabetes management and the occurrence of DRPs should be considered.

Thank you for underlining this deficiency. We have now added implications of these lifestyle factors in the implications for clinical practice and policy development section. [Addressed in Line numbers 324-327]

When presenting the results related to DRPs, provide more context on the significance of specific drug classes and medications implicated in causing DRPs. Additionally, discuss any potential implications for clinical practice and patient care.

We have now added context in the implications for clinical practice and policy development section. [Addressed in Line numbers 322-334]

The association analysis between DRPs and sociodemographic variables is informative, but the interpretation could be enhanced by discussing possible reasons for the observed associations or lack thereof. For instance, why might the duration of diabetes and fruit serving be associated with DRPs while other variables are not?

Thank you highlighting this deficiency. We have now added the context relating to this. [Addressed in Line numbers 255-256, 260-263, 267-269, 301-308]

In the discussion section, build upon the results by synthesizing findings from the literature and offering insights into the broader implications for diabetes management and pharmaceutical care practices in similar settings.

We have highlighted this section in the implications. [Addressed in Line numbers 312-321, 327-334]

Overall, the results section provides valuable data but could benefit from deeper analysis and interpretation to enhance the understanding of the findings and their relevance to the study objectives.

Thank you for the general positive assessment of the paper. We have now tried to interpret within our discussion section. [Addressed in Line numbers 255-256, 260-263, 267-269, 301-308]

The statistical analysis performed in this study appears to be appropriate for the research objectives outlined. However, there are a few areas where additional clarification or depth could enhance the interpretation of the findings.

We have now tried to explain the statistical clarification in our results in the discussion section. [Addressed in Line numbers 279-286, 300-308]

Consider providing more information about the statistical tests used, particularly regarding the choice of the chi-square test for analyzing associations between categorical variables. Explaining why this test was chosen and whether its assumptions were met would be beneficial.

Thank you for the comment. We have included this information in the manuscript. [[Addressed in Line numbers 139-141]

Provide details on the significance level (alpha) used for hypothesis testing. Mentioning the significance level (typically set at 0.05) would help readers understand the criteria for determining statistical significance.

Thank you for the comment. We have included this in our manuscript. [Addressed in Line numbers 141]

When discussing the results of the chi-square tests, consider reporting the effect sizes, such as Cramer's V or Phi coefficient, in addition to p-values. Effect sizes provide valuable information about the strength and practical significance of the associations observed.

Thank you for the comment. With due respect we want to inform the reviewer that we did not report the effect sizes, such as Cramer's V or Phi coefficient, in addition to p-values. We have included this in our limitations. [Addressed in Line numbers 344-346]

Evaluate the statistical tests' power to ensure that the sample size was sufficient to detect meaningful associations. If the study had limited power, acknowledge this limitation and discuss its potential impact on the interpretation of the results.

Thank you for the comment, we have included this in our limitation section. [Addressed in Line numbers 346-347]

Discuss any potential confounding variables controlled for in the analysis or factors considered but not included in the final models. This information would help readers understand the robustness of the findings and the extent to which alternative explanations were ruled out.

Variables such as financial aspect, psychological aspect were not measured that could have influenced the cause and effect. We have included this in our limitation section. [Addressed in Line numbers 338-340]

Consider performing subgroup analyses or stratifying the results by relevant variables to explore potential effect modifiers or identify subgroup differences in the associations examined.

Thank you for the comment. With due respect we want to inform the reviewer that we did not perform subgroup analyses or stratifying the results by relevant variables to explore potential effect modifiers.

If applicable, discuss any adjustments made for multiple comparisons to control the overall Type I error rate. This is particularly important if multiple comparisons were conducted without appropriate adjustments, as it could inflate the likelihood of false positives.

Thank you for the comment. With due respect we want to inform the reviewer that we did not make out for any such adjustments made for multiple comparisons to control the overall Type I error rate.

Provide information on missing data handling and any sensitivity analyses conducted to assess the robustness of the findings in the presence of missing data or outliers.

No sensitivity analyses were conducted to assess the robustness of findings in the presence of missing data or outliers.

Finally, the clinical significance of the statistically significant associations observed will be discussed. Consider whether the magnitude of the associations is meaningful in practice and how these 

---

## [Decision Letter · Decision Letter 1]

2 Jul 2024

PONE-D-23-23137R1Drug-related Problems in Type 2 Diabetic Patients in Sunwal Municipality, Western NepalPLOS ONE

Dear Dr. Dangi,

Thank you for submitting your manuscript to PLOS ONE. After careful consideration, we feel that it has merit but does not fully meet PLOS ONE’s publication criteria as it currently stands. Therefore, we invite you to submit a revised version of the manuscript that addresses the points raised during the review process.

We look forward to receiving your revised manuscript.

Kind regards,

Naeem Mubarak, PhD

Academic Editor

PLOS ONE

**Additional Editor Comments:**

The manuscript has been critically reviewed and must undergo major revisions to improve its quality

Reviewers' comments:

Reviewer's Responses to Questions

**Comments to the Author**

1. If the authors have adequately addressed your comments raised in a previous round of review and you feel that this manuscript is now acceptable for publication, you may indicate that here to bypass the “Comments to the Author” section, enter your conflict of interest statement in the “Confidential to Editor” section, and submit your "Accept" recommendation.

Reviewer #2: (No Response)

2. Is the manuscript technically sound, and do the data support the conclusions?

Reviewer #2: Partly

3. Has the statistical analysis been performed appropriately and rigorously? 

Reviewer #2: Yes

4. Have the authors made all data underlying the findings in their manuscript fully available?

Reviewer #2: Yes

5. Is the manuscript presented in an intelligible fashion and written in standard English?

Reviewer #2: No

6. Review Comments to the Author

Reviewer #2: Authors have made significant modifications. There are some issues in the revised manuscript that need clarification from the authors.

Please find them below:

1.Anatomical system is still not defined appropriately. Authors are advised to go through the WHO-ATC classification and quote the system as mentioned in the WHO-ATC system. For example: It should be Cardiovascular system and genitourinary system and sex hormones. The alimentary canal is still mentioned at certain points in the text.

2.‘Major Therapeutic Class of Drugs Used by Patients’: Authors have not provided the denominator (total number of drugs) from which percentages are calculated.

3.Duration of diabetes is also inadequately described. The sum of individuals is not matching with 182.

4. PCNE C7.1‘Patient doesnot take drug at all and C8 Non-compliance, I understand , they are interrelated. But to bring clarity to readers, authors should mention to which group they categorized the Patients not adhering to drug therapy. Was it considered under C7.1 or C8

5.The present study did not find any association of DRPS with gender. Hence authors should remove the statement ‘ Among the 97 patients with DRPs, the occurrence of DRP was more common among females’ written in the Discussion section.

6.The discussion part on fruit consumption is wrongly interpreted by the authors. DRPs were more common in the group eating less servings of fruits.

7.A significant % of included individuals were older adults (50-60 years). Authors have still not mentioned the fact that they didn't observe adverse drug reactions as the common DRPs. ADRs are the commonest DRPs. Though authors mention in lines 274-276 that they did not observe any association of DRPs with age, they should specifically mention that ADRs were not a common observation and lend a suitable explanation for the same.

8.In Abstract: Authors mention the biguanides and sulfonylureas to be responsible for the DRPs. Just because these were the common classes prescribed doesnot mean they are associated with DRPs as head to head comparisons between the rates of DRPs with the two classes are not described by the paper. Rather, authors should mention biguanides and sulfonylureas were the common classes of drugs prescribed.

9.Likewise, authors should state Drug classes associated with DRPs and not causing DRPs in Tables such as Table 4

10.‘Diabetic’ word can be removed from the sentences where ‘patients with type 2 diabetes’ are mentioned

11.The entire text needs Grammar correction.

7. PLOS authors have the option to publish the peer review history of their article (what does this mean?). If published, this will include your full peer review and any attached files.

Reviewer #2: No

---

## [Author Response · Author response to Decision Letter 1]

30 Jul 2024

Review Comments to the Author

Reviewer #2: Authors have made significant modifications. There are some issues in the revised manuscript that need clarification from the authors.

Please find them below:

1. Anatomical system is still not defined appropriately. Authors are advised to go through the WHO-ATC classification and quote the system as mentioned in the WHO-ATC system. For example: It should be Cardiovascular system and genitourinary system and sex hormones. The alimentary canal is still mentioned at certain points in the text.

Response to reviewer: We have now defined the term and modified the text and table as suggested. We have used the reference (https://atcddd.fhi.no/atc_ddd_index/) to make the ATC classification for the therapeutic classification of drugs used by patients which totally aligns with the reviewer’s comments too. [Addressed in Line numbers 178-182] 

2.‘Major Therapeutic Class of Drugs Used by Patients’: Authors have not provided the denominator (total number of drugs) from which percentages are calculated.

Response to reviewer: We have now mentioned the total number of drugs (i.e. n=679) from which the percentages are calculated. [Addressed as in Table 2]

3. Duration of diabetes is also inadequately described. The sum of individuals is not matching with 182.

Response to reviewer: We have now added the meaning for the duration of diabetes. We meant it for how long the patient had diabetes. It was divided into recently diagnosed, (who were recently diagnosed as per laboratory confirmation) and patients with diabetes for less than 2 years, 2-5 years, 5-10 years and more than 10 years. The sum is 182. [Addressed in Line numbers 229-231]

4. PCNE C7.1‘Patient does not take drug at all and C8 Non-compliance, I understand, they are interrelated. But to bring clarity to readers, authors should mention to which group they categorized the Patients not adhering to drug therapy. Was it considered under C7.1 or C8

Response to reviewer: Thank you for your comment. We categorized patients who are not taking their medication without any apparent reason under category C7.1. Those who are not taking their medication due to specific reasons, such as the cost of treatment, lack of information about administration, forgetting to take the medication, demographic factors (e.g., unavailability of medical services in rural areas), side effects of drugs, etc., were categorized under C8.

After considering your comments, we discussed this categorization in detail and concluded that regardless of the reasons or conditions, not taking or taking less medication is related to patient behaviour and should be classified under C7. Since both categories (C7.1 and C8) represent aspects of non-compliance, and to avoid confusion for readers, we decided to merge these two categories. We have now combined the values for C7.1 and C8 under a single category, C7.1, and have updated the relevant sections in the table, results, discussion, and metadata accordingly.

5. The present study did not find any association of DRPs with gender. Hence authors should remove the statement ‘Among the 97 patients with DRPs, the occurrence of DRP was more common among females’ written in the Discussion section.

Response to reviewer: Thank you for the suggestion. We have now removed this sentence from the discussions section. [Addressed in Line numbers 288-289].

6. The discussion part on fruit consumption is wrongly interpreted by the authors. DRPs were more common in the group eating less servings of fruits.

Response to reviewer: Thank you for the comment, We have now corrected this interpretation. [Addressed in line number 326-329.

7. A significant % of included individuals were older adults (50-60 years). Authors have still not mentioned the fact that they didn't observe adverse drug reactions as the common DRPs. ADRs are the commonest DRPs. Though authors mention in lines 274-276 that they did not observe any association of DRPs with age, they should specifically mention that ADRs were not a common observation and lend a suitable explanation for the same.

Response to reviewer: Thank you for underlining this deficiency. We have now addressed this issue. [Addressed in Line numbers 293-298].

8. In Abstract: Authors mention the biguanides and sulfonylureas to be responsible for the DRPs. Just because these were the common classes prescribed does not mean they are associated with DRPs as head-to-head comparisons between the rates of DRPs with the two classes are not described by the paper. Rather, authors should mention biguanides and sulfonylureas were the common classes of drugs prescribed.

Response to reviewer: Thank you for highlighting this deficiency and suggestion. We have now added the sentence in the abstract section. [Addressed in Line number 43]

9. Likewise, authors should state Drug classes associated with DRPs and not causing DRPs in Tables such as Table 4

Response to reviewer: Thank you for highlighting this and your suggestion. We have now changed the “Drug classes associated with DRPs and not causing DRPs. We have replaced the word ‘drug causing’ with drugs associated too. [Addressed in Table 4]

10. ‘Diabetic’ word can be removed from the sentences where ‘patients with type 2 diabetes’ are mentioned

Response to reviewer: We have now removed the word diabetic from the sentence where ‘patient with type 2 diabetes’ is mentioned. [Line 36]

11. The entire text needs Grammar correction

Response to reviewer: Thank you for your feedback regarding the grammar in the manuscript. We have now utilized the services of a professional English language editing service to correct the grammatical errors throughout the manuscript. They have also assured us that they are ready to re-edit the text if any further language changes are required to meet the journal's standards.

We have restructured some of the sentences, Table 1 and Table 3.

---

## [Decision Letter · Decision Letter 2]

20 Aug 2024

Drug-related Problems among Type 2 Diabetic Patients in Sunwal Municipality of Western Nepal

PONE-D-23-23137R2

Dear Dr. Nim Bahadur Dangi,

We’re pleased to inform you that your manuscript has been judged scientifically suitable for publication and will be formally accepted for publication once it meets all outstanding technical requirements.

Kind regards,

Naeem Mubarak, PhD

Academic Editor

PLOS ONE

Additional Editor Comments (optional):

The manuscript requires no further revisions.

Reviewers' comments:

Reviewer's Responses to Questions

**Comments to the Author**

1. If the authors have adequately addressed your comments raised in a previous round of review and you feel that this manuscript is now acceptable for publication, you may indicate that here to bypass the “Comments to the Author” section, enter your conflict of interest statement in the “Confidential to Editor” section, and submit your "Accept" recommendation.

Reviewer #2: All comments have been addressed

Reviewer #3: All comments have been addressed

2. Is the manuscript technically sound, and do the data support the conclusions?

Reviewer #2: Yes

Reviewer #3: Yes

3. Has the statistical analysis been performed appropriately and rigorously? 

Reviewer #2: Yes

Reviewer #3: Yes

4. Have the authors made all data underlying the findings in their manuscript fully available?

Reviewer #2: Yes

Reviewer #3: Yes

5. Is the manuscript presented in an intelligible fashion and written in standard English?

Reviewer #2: Yes

Reviewer #3: Yes

6. Review Comments to the Author

Reviewer #2: The authors have made significant modifications in the manuscript. Discussion section is still lengthy and can be shortened to make it easy to read.

Reviewer #3: The authors have modified the manuscript according to reviewers's recommendation. No further changes are necessary.

7. PLOS authors have the option to publish the peer review history of their article (what does this mean?). If published, this will include your full peer review and any attached files.

Reviewer #2: No

Reviewer #3: No

---

## [Editor Report · Acceptance letter]

28 Aug 2024

PONE-D-23-23137R2 

PLOS ONE

Dear Dr. Dangi, 

I'm pleased to inform you that your manuscript has been deemed suitable for publication in PLOS ONE. Congratulations! Your manuscript is now being handed over to our production team.

Kind regards, 

on behalf of

Dr Naeem Mubarak 

Academic Editor

PLOS ONE